# Simulating Landslide Generated Tsunamis in Palu Bay, Sulawesi, Indonesia

Alexey Androsov * , Sven Harig and Natalja Rakowsky

Alfred Wegener Institute, Helmholtz Center for Polar and Marine Research, Am Handelshafen 12, 27570 Bremerhaven, Germany
* Correspondence: alexey.androsov@awi.de

**Abstract:** The results of an extensive series of numerical experiments of the GNOM-LS model for modelling the physical and energy characteristics of tsunami waves generated by landslides are presented. Based on the published data on the tsunami on 28 September 2018 in Palu Bay, we analysed the sensitivity of the distribution of wave heights along the coastline formed by the landslide system, depending on the characteristics of these landslides and model parameters. The complexity of the work lies in the lack of a holistic picture of the initial information about landslides, their number and accurate measurement data on the height of the waves of the event. We attempted to restore these conditions by comparing numerical simulations for various initialisations of the landslide system with available observational data. It is revealed that the simulated system has a very high sensitivity to the initial conditions and characteristics of landslides. An essential task of the work is interpreting a complex picture of the nonlinear interaction of tsunami waves with minor changes in the initial characteristics of landslides. Based on the numerical simulation of single landslides and a complete system of landslides, an analysis of the complex structure of the nonlinear interaction of tsunami waves is carried out.

**Keywords:** landslide; tsunami; numerical model; wave interaction; nonlinearity; energy characteristics; spectral analysis





## 1. Introduction

Coastal damage caused by a landslide occurs when the adhesion force of rock elements is inferior to the gravitational force of separation of part of the coastal rock. Rapid climate change has the potential to impact the preconditioning and triggering factors that were suggested as possible contributors to the initiation of submarine landslides: hurricanes or cyclic loading [1], rainfall [2,3], gas hydrate dissociation [4], sea-level change [5], and some others [6]. Human activities can also contribute to submarine landslide triggering. However, the most common mechanism for triggering multi-mega landslides is earthquakes' destabilisation of prone material [3,7,8].

On 28 September 2018, a strong earthquake of MW 7.5 hit the northwestern part of Sulawesi, Indonesia. Despite the fault mechanism being predominantly strike-slip, large regional tsunamis were generated, primarily affecting the Palu Bay area, located 50 km southeast of the epicentre. Palu City, located at the end of the bay, experienced enormous devastation by the earthquake, tsunami inundation, and soil liquefaction less than 5 min after the event.

The observations suggest that submarine landslides contributed considerably to the tsunami. In surveys after the event, conducted by various organisations based on video observations and satellite images [9–12], field survey observations [13–15], analysis of the tide gauges [16,17], and numerical simulations [11,16–18], the impact of the waves as well as the coastal subsidence was investigated. Several locations of potential landslides in the bay area were identified.

Three main directions of landslide hydrodynamics can be distinguished. The simplest is the landslide model set by its centre of gravity. It is a non-stationary point moving along a given trajectory when setting some unchanged characteristics of the landslides—thickness, mass, and volume. Nonlinear dispersion equations describe the resulting surface waves [19–22].

The following extensive type of models of landslide mechanisms is based on the representation of the lower layer as a granular flow, taking into account inter-granular stresses regulated by Coulomb friction [23–26].

Another approach to simulating landslide-induced tsunamis is based on solving the Navier-Stokes (NS) equations with a simplified definition of the density and viscosity field. The derived velocity values are used in two advection equations concerning variables expressing air/landslide material and water/landslide material fractions in the elementary volume. This method allows a simple way to determine the position of the interfaces of the three-phase medium landslide-water-air [27–34]. Note that in the work of [25], it was demonstrated that in comparing the Newtonian fluid model and the granular material model, the difference in tsunami waves is observed only near the landslide source.

The reproduction of the landslide dynamics and its consequences based on the solution of a boundary value problem in an arbitrary domain should include both detailed data on the topography and bathymetry of the region and information on the characteristics of the landslide itself: dimensions, density, porosity, slope angle, frontal geometry, and velocity [32]. In most cases, such information about a landslide is based only on visual observations and incomplete data [18,35–37]. The lack of necessary information leads to significant inaccuracies in modelling the wave propagation field for various forms of nonlinear dispersion interaction [38,39].

As seen from the literature review above, for the 2018 Sulawesi event, only fragmented information is available on the landslides, their number, and observations in tsunami-affected areas. Based on this information, attempts have already been made to model landslide mechanisms in Palu Bay. Based on the results of numerical modelling of Delft 3D Flow, Ref. [18] performed a detailed analysis of the direction of wave propagation and arrival time of the first tsunami wave in Palu City from one of the landslide sources. The authors note that other landslide sources could generate subsequent waves, and their accurate modelling will require detailed bathymetric data. Ref. [16] discusses a simplified depth-averaged two-dimensional model as the basis for modelling two landslide sources. One is located in the north of the bay, the other in the southern part. The model shows good alignment with the amplitude of the running wave on the west coast but poorly predicts the running wave's time on the bay's east coast.

The purpose of this work is not only an attempt to reconstruct tsunami waves in Palu Bay under the influence of a landslide mechanism using a numerical model but mostly an analysis of the complex picture of the nonlinear interaction of these waves. The reproduction of landslide dynamics is based on the search for an optimal solution by comparing 22 local observations of wave height with model calculations of six landslide sources. The selected scenarios include the analysis of various physical characteristics of landslides (size, density and response time), as well as various parameters of the numerical model.

A landslide model GNOM-LS based on the NS equations for the two-layer boundary-value problem landslide-water-air in an arbitrary domain is considered in detail in the paper. The problem is formulated in Cartesian coordinates and transformed into curvilinear boundary coordinates. The numerical method is realised by splitting in coordinate directions.

The following Section sets a two-layer boundary task in the area so that the landslide, given by the size and average density, fills the lower layer. The numerical method uses a switcher to approximate advection to third-order accuracy, a TVD procedure (Total Variation Diminishing), a wetting and drying algorithm, and algorithms for layer friction parameterisation. The computation and analysis of the scenarios for multi-slide modelling are given in Section 3. Section 4 analyses the nonlinear effects of tsunami wave interaction

and each landslide's contribution to the flood zone. Section 5 discusses the results of the work and concludes.

## 2. Landslide Model GNOM-LS

### 2.1. Governing Equations

Let the undisturbed surface of the water coincide with the horizontal plane $X0Y$ of the right Cartesian coordinate system. The $0Z$ axis is directed vertically upwards. In the region $Q_T = Q \times [0, T]$ where $Q = \{x, y; x, y \subset \Omega\}$ is the domain bounded by the free surface of the water $\zeta(x, y, t)$, the bottom $h_*(x, y)$, and the side surface $\partial Q$, $0 \le t \le T$. Consider a uniform layer of liquid with density $\rho_1$, thickness $h_1(x, y)$, and free surface $\zeta_1(x, y, t)$, over which there is a uniform layer of liquid with density $\rho_2$, with the depth of the undisturbed surface thickness $h_2(x, y)$, and free surface $\zeta$ (see Figure 1). In this notation, $h_1(x, y) = h_*(x, y) - h_2(x, y)$.

Denote the average flow velocity vectors in the lower and upper layers, $\mathbf{u}_1 = (u_1, v_1)$ and $\mathbf{u}_2 = (u_2, v_2)$ respectively, and record the motion equations for the layers:

$$\frac{\partial \mathbf{u}_j}{\partial t} + (\mathbf{u}_j \cdot \nabla)\mathbf{u}_j + \frac{1}{\rho_j}\nabla p_j + f\mathbf{u}_j^* = \mathbf{D}_j, \tag{1}$$

where $j = 1, 2$, $j = 1$—landslide, $j = 2$—water layer, $\nabla = (\partial/\partial x, \partial/\partial y)$, $f$ is the Coriolis parameter, $\mathbf{u}_j^* = (-v_j, u_j)$, $p_j$—pressure, a definition for each layer will be given below. For dissipate terms $\mathbf{D}_j$, we take:

$$\mathbf{D}_j = (\tau_{j+1} - \tau_j)/(H_j\rho_j) + K_j\nabla^2 \cdot \mathbf{u}_j, \quad \tau_j = C_j(\mathbf{u}_j - \mathbf{u}_{j-1})\sqrt{(u_j - u_{j-1})^2 + (v_j - v_{j-1})^2}.$$

$C_j$, $K_j$—coefficient of layer friction and horizontal viscosity, respectively, $\mathbf{u}_0 = 0$ - velocity at the height of the bottom roughness and $H_j$—full-thickness of each layer.

Equation of continuity for layers:

$$\frac{\partial H_j}{\partial t} + \nabla(H_j\mathbf{u}_j) = 0, \tag{2}$$

$H_1 = h_1 + \zeta_1$, $H_2 = h_2 + \zeta - \zeta_1$.

For the pressure gradient for each layer, we have:

$$\frac{1}{\rho_1}\nabla p_1 = g_1'\nabla\zeta_1 + g_2'\nabla\zeta, \quad \frac{1}{\rho_2}\nabla p_2 = g_2'\nabla\zeta.$$

Here $g_j' = g(\rho_j - \rho_{j+1})/\rho_j$—reduced gravitational constant, $g$—gravitational constant and $\rho_3 = \rho_a$—atmospheric pressure.

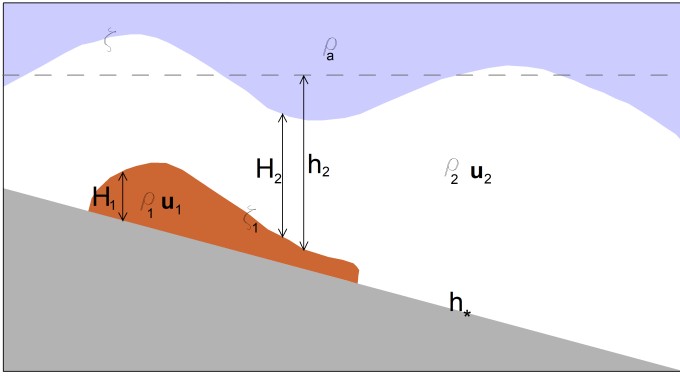

**Figure 1.** Schematic plot of the two-layer system. Index 1 depicts the landslide, 2—water and a—atmosphere.

The landslide model demands a strong horizontal viscosity in places of large gradients of the solution. Large gradients may form due to a nonlinear steepening of the wavefront or on reflections from jumps in the topography or coast. Using uniform horizontal viscosity on non-uniform grids is a lousy choice leading to substantial time-stepping limitations. Using the coefficient depending on the grid size proved inefficient, too, as one needs high viscosity only when large velocity gradients are observed. For this reason, the coefficient of horizontal viscosity was determined according to the Smagorinsky parameterisation [40]:

$$K_j = A_s \Delta_x \Delta_y \sqrt{u_{j_x}^2 + v_{j_y}^2 + \frac{1}{2}(u_{j_y} + v_{j_x})^2},$$

In numerical experiments, we use a coefficient of $A_s = 0.1$, ensuring the solution's stability.

### 2.2. Curvilinear Coordinate System

We introduce horizontal curvilinear coordinates fitted to the shape of domain $\Omega(x, y)$. Let us consider the transformation:

$$\xi = \xi(x, y), \eta = \eta(x, y),$$

with Jacobian $I = \partial(\xi, \eta)/\partial(x, y) = J^{-1}$, $J = x_\xi y_\eta - x_\eta y_\xi$, $0 \neq J < \infty$ and with basis vectors: $\mathbf{e}_i = \mathbf{r}_{\xi^i}$, $\mathbf{e}^i = \nabla \xi^i$; $\xi^1 = \xi$, $\xi^2 = \eta$, $\mathbf{r} = (x, y)$. With an appropriate choice of two pairwise-opposite segments of a side surface $\partial \Omega$, the domain $\Omega$ is mapped into a rectangle $\Omega_*(\xi, \eta)$ with the outline $\partial \Omega_*$.

Let us write Equations (1) and (2) in new coordinates relative to the vector $\boldsymbol{w}_j = (u_j, v_j, H_j)$, omitting, for ease of notation, the layer index $j$:

$$\partial_t \boldsymbol{w} + A \partial_\xi \boldsymbol{w} + B \partial_\eta \boldsymbol{w} = \boldsymbol{\psi}. \tag{3}$$

where

$$A = \begin{pmatrix} U & 0 & H\xi_x \\ 0 & U & H\xi_y \\ g'\xi_x & g'\xi_y & U \end{pmatrix}, \quad B = \begin{pmatrix} V & 0 & H\eta_x \\ 0 & V & H\eta_y \\ g'\eta_x & g'\eta_y & V \end{pmatrix}, \quad \boldsymbol{\psi} = (\boldsymbol{\phi}, 0).$$

$\boldsymbol{\phi} = (\mathbf{D} + g'\nabla\zeta)$ and $U = \mathbf{u}\nabla\xi$, $V = \mathbf{u}\nabla\eta$—contravariant components of the velocity.

### 2.3. Numerical Realisation

The problem is solved on a uniform rectangular grid $Q_\Delta^*$ in the rectangle $Q^*$, which is a map of a 2-D curvilinear grid $Q_\Delta$ generated in a physical domain $Q$. A curvilinear grid $\Omega_\Delta$ in the domain $\Omega$ is constructed using an elliptical method [41] with orthogonalisation at the boundary $\partial \Omega$.

The splitting scheme in two directions is the most convenient scheme for solving Equation (3) for the vector $\boldsymbol{w}$:

$$(I + \frac{\kappa}{2} A \delta_\xi)\boldsymbol{w}^* = (I - \frac{\kappa}{2} B \delta_\eta)\boldsymbol{w}^k + \frac{\Delta t}{2} \boldsymbol{\psi}^k,$$

$$(I + \frac{\kappa}{2} B \delta_\xi)\boldsymbol{w}^{k+1} = (I - \frac{\kappa}{2} A \delta_\eta)\boldsymbol{w}^* + \frac{\Delta t}{2} \boldsymbol{\psi}^*,$$

where $\kappa = \Delta t / \Delta$; $\Delta = \Delta \xi = \Delta \eta$, $k = 0, 1, 2, \ldots, [T/\Delta t]$, $\delta_\xi$, $\delta_\eta$—central difference operators; $I$—identity operator.

The structure of the matrix $A$ allows excluding $\zeta^*$ from the first and third equations for $\boldsymbol{w}^*$ and to obtain a solution for $U^*$ that is solved by the effective three-point Thomas algorithm with boundary conditions for $U$ component of the velocity. The unknown variables $\zeta^*$, $V^*$ are now explicitly defined. Similarly, excluding $\zeta^{k+1}$ from the second and third equations for $\boldsymbol{w}^*$, we come to the edge problem for $V^{k+1}$, also solved by the three-point Thomas algorithm for $V$ component of the velocity. The functions $\zeta^{k+1}$ and

$U^{k+1}$ are then explicitly defined. A similar structure is employed for the algorithm for the lower layer.

The numerical scheme implemented in the model has the second order of accuracy in time and space.

### 2.3.1. Wetting and Drying Algorithm

Modelling the wetting and drying processes is crucial in generating tsunamis by landslides. Various approaches to constructing the drying procedure should be built into the general algorithm at points where a layer degenerates. In this model, we use two different robust approaches. The essence of the first method is to establish a barrier for the flow if the total thickness of the layer in the computational point becomes less than some given threshold $H_{min}$. The thickness of the layer $H_j$ is determined in nodes with indexes $(m + 1/2, n)$, while the values of $V$, at the upper side of rectangle $Q^*$, are regarded in the nodes $(m + 1/2, n + 1/2)$. We assume that the total thickness of the layer in this point fulfils $H_{j(m+1/2,n+1/2)} = H_c = (H_{j_n} + H_{j_{n+1}})_{m+1/2}$, for points $H_{j(m+1/2,n-1/2)} = H_m = (H_{j_n} + H_{j_{n-1}})_{m+1/2}$ and $H_{j(m+1/2,n+3/2)} = H_p = (H_{j_{n+1}} + H_{j_{n-2}})_{m+1/2}$. The requirement of the impenetrability of the flow through a side of the cell is expressed by the conditions:

$$V_{m+1/2,n+1/2} = V_c = 0, \text{ if } H_m, H_p \leq H_{min},$$
$$V_c = 0, \text{ if } H_m \leq H_{min}; H_p > H_{min} \text{ and } V_{m+1/2,n+3/2} > 0,$$
$$V_c = 0, \text{ if } H_p \leq H_{min}; H_m > H_{min} \text{ and } V_{m+1/2,n-1/2} < 0.$$

In the other case, the barrier on the side is removed. A similar procedure for the $U$ component of velocity is applied. This procedure permits the inspection of the solution at a small thickness and does not allow a negative depth in the nodes of velocities and layer thickness.

The second method is based on the layer thickness analysis and, depending on it, reduces the equations of motion. At each time step, a spatial mask of the dry, wet, and transitional nodes of the mesh in each layer is defined:

$$
\begin{aligned}
a_{wd} &= 1 & \text{if} & \quad H_j > H_{crit} \\
a_{wd} &= \frac{H_j - H_{min}}{H_{crit} - H_{min}} & \text{if} & \quad H_{min} \leq H_j \leq H_{crit} \\
a_{wd} &= 0 & \text{if} & \quad H_j < H_{min},
\end{aligned}
$$

where $H_{min} < H_{crit}$. After that, the momentum Equations (1) in the layers transform into a balance between the time derivative and the surface slope terms when $H_j \implies H_{crit}$ [42]:

$$\frac{\partial \mathbf{u}_j}{\partial t} + \frac{1}{\rho_j} \nabla p_j + a_{wd}((\mathbf{u}_j \cdot \nabla)\mathbf{u}_j + f\mathbf{u}_j^* - \mathbf{D}_j) = 0.$$

### 2.3.2. Advection

The MUSCL algorithm [43] approximates advection with a switch that allows the use of first, second or third-order accuracy schemes with an attached TVD procedure that controls the behaviour of the solution in regions with sharp gradients and in places where the layers degenerate. Moreover, for simplicity, we will omit the layer index $j$ when deriving the equations since the algorithm is applied in each layer.

Let $L^{\xi^i}$ be a one-dimensional advective operator in the coordinate direction $\xi^i$. A chain of operators $L = L^\sigma L^\eta$ represents an advective transport in an time interval $[k, k+1]$ whose result is denoted by $U^{i^*} = L^\xi L^\eta U^k$. Consider the advection equation in divergent form for one direction:

$$\frac{\partial}{\partial t} J\mathbf{u} + \frac{\partial}{\partial \xi}\phi = 0,$$

where $\phi = U\omega$, $\omega = J\mathbf{u}$ and write the difference approximation of this equation:

$$\omega^{k+1} = \kappa\alpha(\phi_{m+1/2} - \phi_{m-1/2})^{k+1} = \omega^k - \kappa(1-\alpha)((\phi_{m+1/2} - \phi_{m-1/2})^k,$$

where $\kappa = \tau/\Delta$; $0 \leq \alpha \leq 1$.

For the upstream difference of the transport $\phi_{m+1/2}$, we can write:

$$\phi_{m+1/2} = \frac{1}{2}(\phi_m + \phi_{m+1}) - \frac{1}{2}(\phi^+ - \phi^-)_{m+1/2},$$

where $\phi^\pm_{m+1/2} = (U^+\omega^- + U^-\omega^+)$, $U^\pm = \frac{1}{2}(U \pm |U|)$.

TVD schemes have higher accuracy in sharp gradients and contain growth limiters of the total variation. To exclude non-physical oscillations of the solution in regions with sharp gradients, the TVD procedure is added to the numerical schemes, ensuring that the solution's region-wide variation on time layers does not increase. Therefore, for transport, we have an expression with TVD correction:

$$\tilde{\omega}^-_{m+1/2} = \omega_m + \{\frac{s}{4}[(1-\gamma s)\delta^- + (1+\gamma s)\delta^+]\}_m,$$

$$\tilde{\omega}^+_{m+1/2} = \omega_{m+1} - \{\frac{s}{4}[(1-\gamma s)\delta^+ + (1+\gamma s)\delta^-]\}_{m+1},$$

where $\delta^- = \omega_m - \omega_{m-1}$, $\delta^+ = \omega_{m+1} - \omega_m$. The choice of the approximation order of the scheme is determined by the parameter: $s = [2\delta^+\delta^- + \epsilon]/[(\delta^+)^2 + (\delta^-)^2 + \epsilon]$, $\epsilon$ -small number ($O(10^{-6})$):

| | | |
|---|---|---|
| $\gamma = \frac{1}{3}$ | Upstream | $O(\tau^3)$ |
| $\gamma = -1$ | Upstream | $O(\tau^2)$ |
| $\gamma = 0$ | Upstream | $O(\tau)$ |
| $\gamma = 1$ | Central difference | $O(\tau)$ |

The implementation of the second operator in the $\eta$ direction is organised similarly.

2.3.3. Layer Friction

Three schemes for determining layer friction can be used in the model. The first is the friction between the layers with a constant coefficient for each of the layers. The second scheme for determining the friction coefficient is based on changing the layer thickness and surface roughness $z_h$ [44]:

$$C = (\ln((0.5H_j + z_h)/z_h)/\kappa)^{-2}.$$

The third parameterisation of the bottom friction is the Manning formula, which provides the ability to change the coefficient during the movement of a solid landslide on a surface with different characteristics of the slope on land and underwater:

$$C = \frac{gn^2}{H_j^{1/3}},$$

with $n = 0.01$ for a smooth slope, $n = 0.015$ for a rough slope, and $n = 0.12$ for the slide after it rushed into the water.

**3. Results**

In this part of the work, based on the results of numerical modelling of the multi-landslide mechanism, various scenarios for the occurrence of a tsunami wave in Palu Bay are analysed. Due to the uncertainty in the initial data and model parameters, three series of experiments were performed to find one scenario, the results of which best agree with the available wave height observations in the coastal area.

In the first series of experiments (BE), by enumeration of various initialisation parameters of landslides (density, volume), the time of the beginning of their movement and forced stop, the internal parameters of the model (schemes associated with the parametrisation of friction between layers and wetting and drying). One experiment is selected based on a minimum error analysis with observational data.

The second series of experiments (CE) is based on the selected experiment from the first group on a more detailed refinement of the landslide parameters and the model in a smaller range. Scenarios with initial opposite movement in time of landslides are also added. Again, the best scenario for minimising the error is selected.

The final series of experiments (FE), for the best scenario (minimum errors with observational data), is repeated several times only with a change in the landslide density. The best scenario will be used to analyse further the generation of tsunami waves, their nonlinear interaction, and the influence of the contribution of each landslide to the overall structure of flooding.

### 3.1. Data and Initial Model Parameters

The long, narrow Palu Bay is located in Central Sulawesi, Indonesia. A complex geometry characterises it: with a bay length of ∼28 km and a maximum width of about ∼8 km, the shallow coastal zone abruptly slopes down to a depth of 750 m in the central part of the bay (see Figure 2 left panel). In the southern part of the bay, there is a highly variable bathymetry in the transverse direction.

The curvilinear mesh contains 121 × 451 nodes with the horizontal size varying between 30 m and 180 m. Bathymetry and topography are provided by BIG (Badan Informasi Geospasial, Indonesia) with a resolution of 200 m. This bathymetry/topography was interpolated onto a computational mesh; the maximum topography in the landslide model was limited to 13.5 m.

Six potential landslides with observed subsidence were localised according to the work of [45]. The boundaries of the landslides were determined according to the land and sea surface images before and immediately after the Sulawesi earthquake [45] and smoothly extended into the water domain. The slides' volumes depend on the local topography and bathymetry (see Table 1). Estimates of the volumes of some landslides located along the east coast and the Palu coast region are given in [16], and for comparison, we put these values in Table 1. Landslides *E* and *F*, located on the west coast, are not included in the simulation by [16]. The scenarios performed in this paper also used landslide volumes close to those given in [16] (Correction Experiments). Still, the comparison with visual data was slightly worse than the baseline scenario.

22 stations (see Figure 2) with observations of inundation depth [46] were selected for comparison with model simulations. The data used as observations were collected between 29 September and 6 October 2018 through visual measurements.

All experiments were performed for 20 min with a time step of 0.05 s.

**Table 1.** The landslide volume $V_{ref}$ is determined from the topography in the observed subsidence location [45] smoothly extended into the water domain.

| Slide ID | Slide Volume ($V_{ref}$), m$^3$ | Slide Volume *, km$^3$; Station Name, [16] |
|:---:|:---:|:---:|
| *A* | 3,976,358 | 0.02; (Off Wani, LSC2) |
| *B* | 5,023,400 | 0.07; (Off Dupa , LSC4) |
| *C* | 3,660,750 | 0.07; (Off KM Hotel, LSC5) |
| *D* | 8,785,640 | 0.07; (Off West Palu, LSC6) |
| *E* | 6,748,037 | ——- |
| *F* | 7,045,744 | ——- |

* The slide volume given in [16] is shown for comparison.

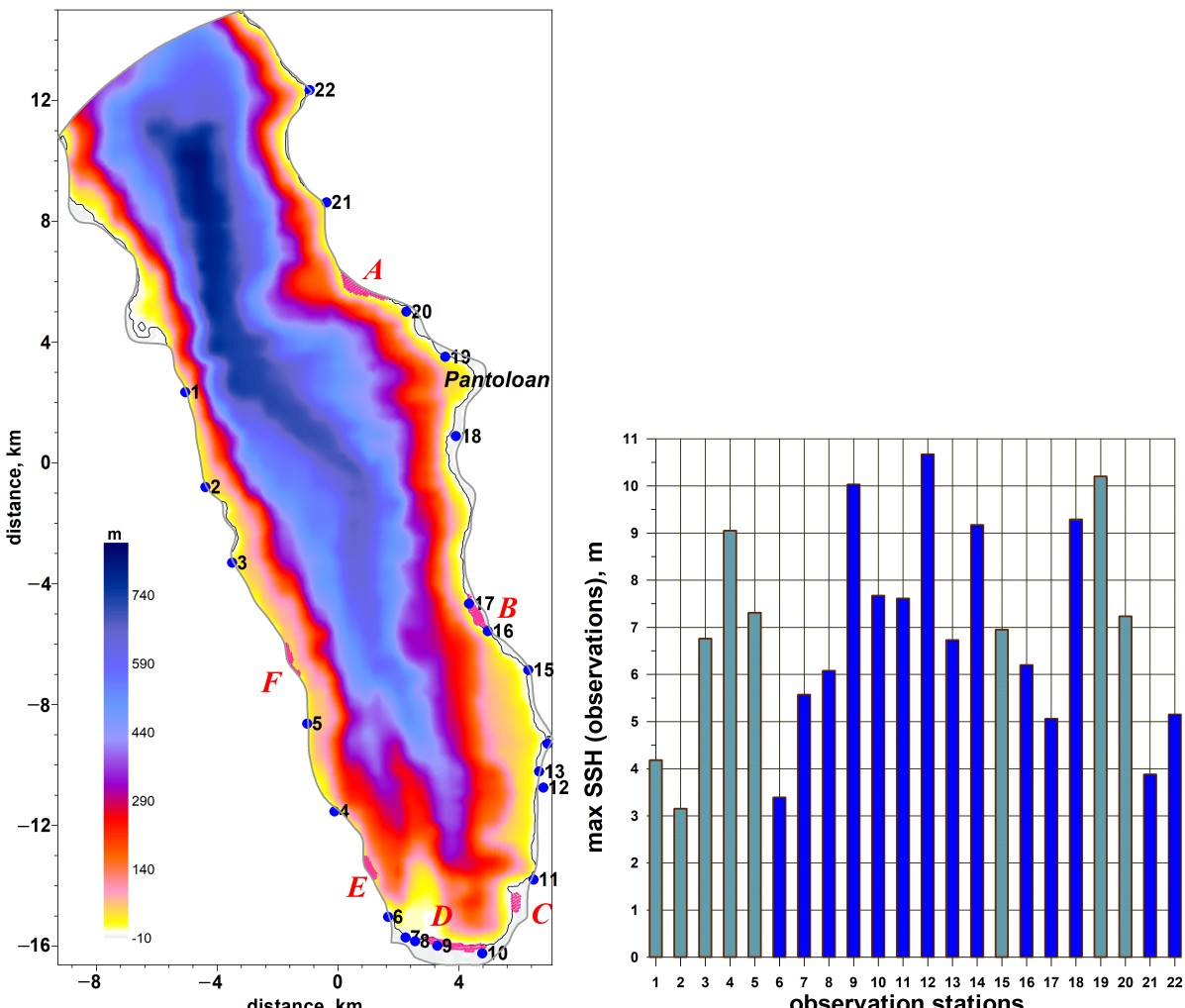

**Figure 2. Left** panel: Model domain, bathymetry, observational locations (1–22), landslide positions ($A - F$). Grey line—topography 10 m; black solid line—0 m depth; **right** panel: observed sea level height for locations 1–22 [46]. Dark blue bars show observation stations located in the coastal zone of the modelling area.

*3.2. Sensitivity Study*

3.2.1. A Detailed Description of the Experiments

Figure 3 shows the results of the analysis of various sets of experiments on identifying initial conditions and model 2 parameters for modelling the multi-landslide in Palu Bay based on a comparison of the maximum wave height at 22 stations of observations and simulations. The estimate is summarised for each experiment for three-time intervals 10, 15 and 20 min calculations for the stations marked in Figure 2.

The basic experiments (BE, Table 2) are set up by varying the initial time of movement of the landslides, the time of their forced stop, and their initial volume. The best agreement with the observations is achieved in the experiment BE-3 over a complete-time cycle. This experiment describes the initial movement of landslides with a simultaneous start (in the counterclockwise direction), without the forced landslide movement stops and with a given initial volume of the slides from Table 1. For experiments BE-1 to BE-18, the maximum wave is formed within 15–20 min after the movement of landslides.

**Table 2.** Characteristic quantities of the basic experiments (BE).

| Number of Basic Experiment | Time Difference between Slides, s ($F \rightarrow A$) | Slide Density [t/m$^3$] | Slide Stopping Time, s | Slide Volume (Table 1) | Layer Friction/Wetting-Drying |
|---|---|---|---|---|---|
| BE-1 | Simultaneously | 2.1 | Never | $V_{ref}$ | Log *./Mask |
| BE-2 | Simultaneously | 2.2 | Never | $V_{ref}$ | Log./Mask |
| BE-3 | Simultaneously | 2.4 | Never | $V_{ref}$ | Log./Mask |
| BE-4 | Simultaneously | 2.4 | 600 | $V_{ref}$ | Log./Mask |
| BE-5 | Simultaneously | 2.4 | 300 | $V_{ref}$ | Log./Mask |
| BE-6 | 5 | 2.4 | 600 | $V_{ref}$ | Log./Mask |
| BE-7 | 10 | 2.4 | 600 | $V_{ref}$ | Log./Mask |
| BE-8 | 15 | 2.4 | 600 | $V_{ref}$ | Log./Mask |
| BE-9 | 20 | 2.4 | 600 | $V_{ref}$ | Log./Mask |
| BE-10 | 15 | 2.4 | 600 | $V_{ref}/8$ | Log./Mask |
| BE-11 | 15 | 2.4 | 300 | $V_{ref}/2$ | Log./Mask |
| BE-12 | 15 | 2.4 | 600 | $V_{ref}/4$ | Log./Mask |
| BE-13 | 15 | 2.7 | 600 | $V_{ref}/8$ | Log./Mask |
| BE-14 | 15 | 2.1 | 600 | $V_{ref}/8$ | Log./Mask |
| BE-15 | 15 | 2.4 | 300 | $V_{ref}/8$ | Log./Mask |
| BE-16 | 15 | 2.4 | 600 | $V_{ref}/8$ | Manning/Mask |
| BE-17 | 15 | 2.4 | 600 | $V_{ref}/8$ | Constant/Mask |
| BE-18 | 15 | 2.4 | 600 | $V_{ref}/8$ | Constant/Barrier |
| BE-19 | 5 | 2.4 | Never | $V_{ref}$ | Constant/Mask |
| BE-20 | 10 | 2.4 | Never | $V_{ref}$ | Constant/Mask |
| BE-21 | 15 | 2.4 | Never | $V_{ref}$ | Constant/Mask |
| BE-22 | 20 | 2.4 | Never | $V_{ref}$ | Constant/Mask |

* Logarithmic.

Based on the initial conditions of experiment BE-3, we conduct several corrective experiments (CE, see Table 3) to study the effect of the time of landslide movement stops and the direction in which they are launched. Variants 1–5 describe the initial counterclockwise start-up, and variants 6–9 are the clockwise alternative with an interval between the start of movement of 5 s. The RMSD error analysis shows that the base experiment BE-3 results in the lowest error. Furthermore, triggering the slides in the clockwise order leads to a significant deterioration in the results at 20 min (see Figure 3 middle panel).

Note that at three observation stations (9, 10 and 18), the tsunami wave does not cause flooding in model simulations for all experiments. We assume this is due to coarse initial topography. In the simulated area, the height above sea level at these points turned out to be 7, 6.5 and 5.2 m, respectively. By excluding these three points from the calculation of RMSD, the total error is reduced by more than 1 m.

**Table 3.** Summary of characteristic quantities of the correction experiments (CE). The slide density is 2.4 t/m$^3$.

| Number of Correction Experiment | Time Difference between Slides, s | Slide Volume, Layer Friction Parametrisation, Wetting-Drying Algorithm | Direction of Slide Movement |
|---|---|---|---|
| CE-1 * | Simultaneously | $V_{ref}$<br>Logarithmic<br>Mask | —— |
| CE-2 | Simultaneously | $V_{ref}$<br>Manning<br>Mask | —— |
| CE-3 | Simultaneously | $V_{ref}$<br>Constant<br>Mask | —— |
| CE-4 | Simultaneously | $V_{ref}$<br>Logarithmic<br>Barrier | —— |
| CE-5 | Simultaneously | $V_{ref}/2$<br>Logarithmic<br>Mask | —— |
| CE-6 | 5 | $V_{ref}$<br>Logarithmic<br>Mask | $A \rightarrow F$ |
| CE-7 | 10 | $V_{ref}$<br>Logarithmic<br>Mask | $A \rightarrow F$ |
| CE-8 | 15 | $V_{ref}$<br>Logarithmic<br>Mask | $A \rightarrow F$ |
| CE-9 | 20 | $V_{ref}$<br>Logarithmic<br>Mask | $A \rightarrow F$ |

* The CE-1 experiment is similar to BE-3.

The final part of the experiments is related to the choice of the density composition of the simulated landslides (see Table 4), based on the base experiment BE-3. The density of the landslides ranges from 2.4 to 1.5 t/m$^3$. In this series of experiments, the baseline scenario again shows the smallest RMSD error in the final phase of the model simulations—20 min. Note also that when the density decreases to 2.0 t/m$^3$, the comparison results diverge by 15 and 20 min. In addition, at the time of 10 min, the RMSD errors for the entire ensemble of experiments slightly differ (see Figure 3 right panel).

**Table 4.** Summary of characteristic quantities of the final experiments (FE).

| Number of Final Experiment | Slide Density, t/m$^3$ |
|---|---|
| FE-1 * | 2.4 |
| FE-2 | 2.0 |
| FE-3 | 1.9 |
| FE-4 | 1.7 |
| FE-5 | 1.5 |

* The FE-1 experiment is similar to BE-3.

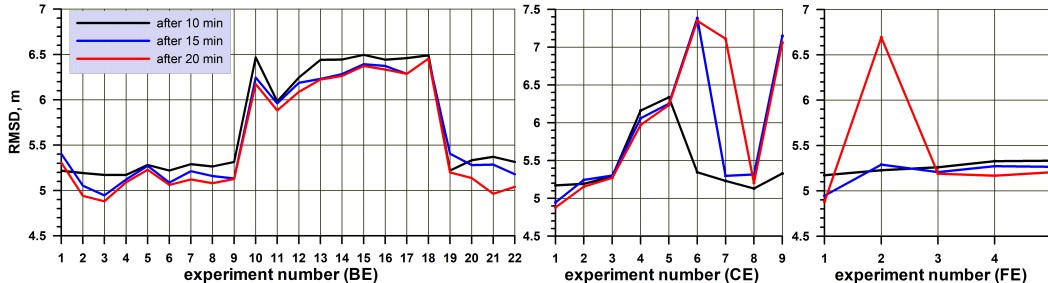

**Figure 3.** Cumulated deviations (RMSD) for 22 coastal observations (Figure 2) for three-time intervals, 10, 15 and 20 min. **Left** panel—basic experiments (BE, Table 2); **middle** panel—correction experiments (CE, Table 3); **right** panel—final experiments (FE, Table 4).

### 3.2.2. Analyse the Results Based on the Best Match Experiment BE-3

Figure 4 shows the spatial evolution of the surface waves over the first 3.5 min of experiment BE-3. The waves propagate in all directions and there is a complex picture of the wave interaction and refraction from the coast. The largest amplitude and the most extensive spatial structure originated by landslide *B*. Note that the volume of this landslide is minor compared to most other landslides except *A* and *C*. The influence of this landslide *B* plays a decisive role in stations 2–17. Its shock wave practically does not change the amplitude while propagating to the opposite shore. The influence of landslides *E*, *D* and *C* due to the location in the shallow zone remains relatively local and determines, together with the wave from landslide *B*, the height of the wave at stations 5–11. The effect of the wave generated by landslide *A* determines the flooding at 1, 20–22 stations. A more detailed analysis of the impact of each of the individual landslides will be done later.

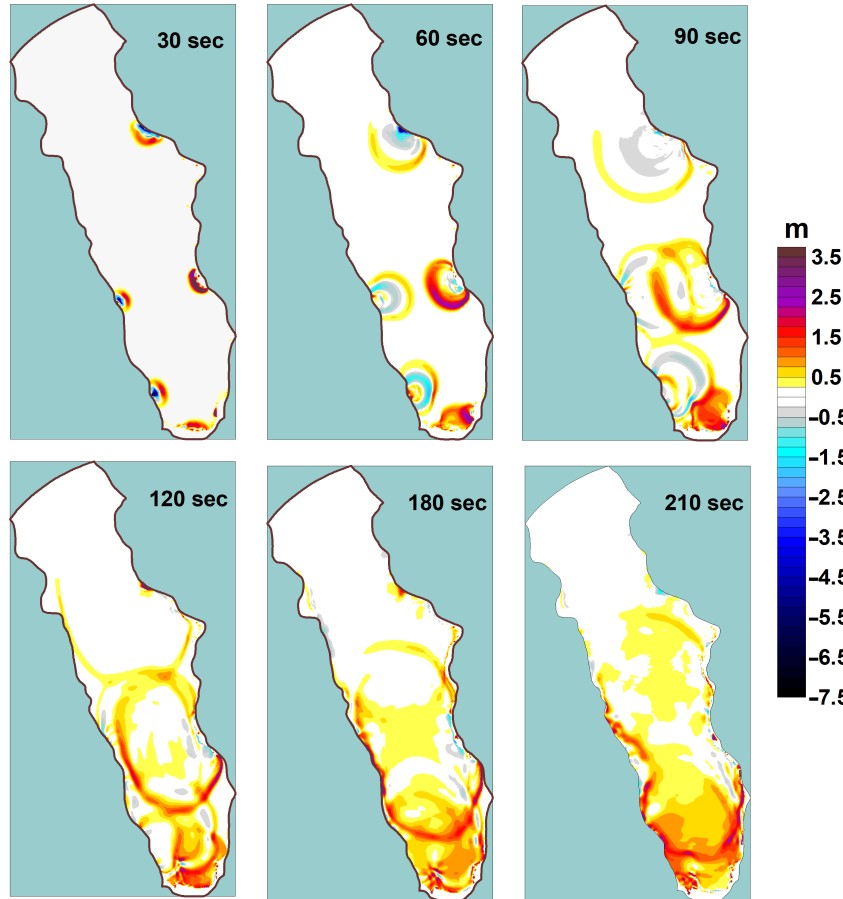

**Figure 4.** Temporal evolution of the basic experiment BE-3 with activation of the six landslides simultaneously.

A comparison of the maximum wave height for 20 min for all model experiments with observation at 22 shore stations is shown in Figure 5. The negative value corresponds to a situation when field observations exceed the model results. As seen from the comparison of the results, the absolute error is at a maximum at stations 12, 13 and 17. In the remaining region, the difference between simulation and observations is almost systematic, slightly improving or worsening depending on the chosen scenario. As noted above, model simulations practically have no flooding at these observation stations. For the correction scenario with the simultaneous start of all landslides (CE-6–CE-9) and the opposite direction of slide movement (clockwise), a significant overestimation of the simulation results at locations 4–5 and 12–13 occurs. Note that these pairs of stations are located on opposite shores of the south part of Palu Bay. Such a substantial excess of the simulated wave may signal an incorrect interaction of the wave generated by landslide *B* and landslides of the southern coast.

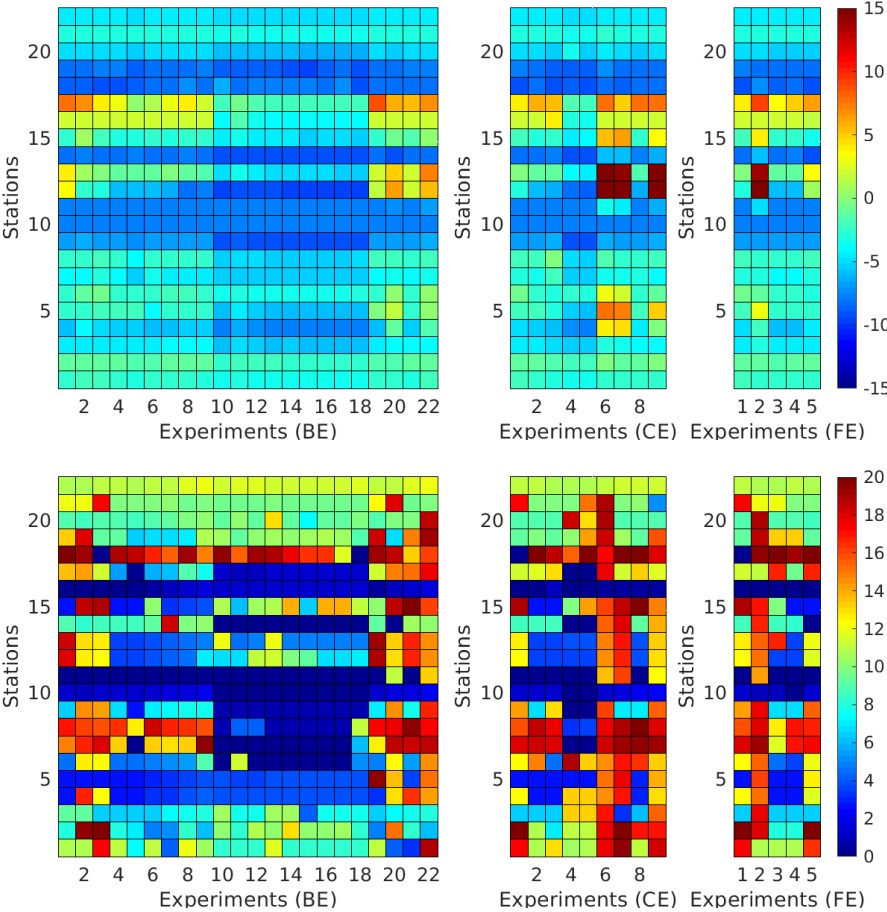

**Figure 5. Upper** panel—the difference between model results after 20 min simulation and observations; **bottom** panel—the time of the maximum wave height of all experiments at observation stations (Figure 2). **Left**—basic experiments (BE); **middle**—correction experiments (CE); **right**—final experiments (FE).

It should also be noted that only 14 out of 22 field measurement stations in the model area lie on the coast due to coarse topography data used in the simulations. In addition, it can be assumed that the errors occurring at the other stations (1–5, 15 and 19–20 Figure 2 right panel) may be associated with the estimate of the wave height in open water compared to the wave height on the shore, based on observations.

Note that in the number of observation points, our best match scenario (BE-3) is underestimated at some points in terms of wave height on the coastal observation stations. This is primarily due to the very coarse initial bathymetry and topography available. Part

of the observation stations on our calculation grid ended up in the open water area of the modelled site (Figure 2). In contrast, all observation stations are located on the coast. Station 19 (Pantoloan) is indicative in this regard. According to observations, the beach's wave height at this point exceeds 10 m. In our modelled area, the station's position lies in the sea and coincides with the position of the tidal station. Observations at this tide gauge station show that the maximum wave height is 1.74 m. The model computation shows almost the same wave height of 2.06 m.

According to model simulations, stations 10 and 11 have no flooding in the coastal zone. Again, we interpret this with a very rough topography database.

In Figure 5, the bottom panel shows the time of occurrence of the maximum wave height at the observation points for all experiments performed. Areas without flooding in model calculations are highlighted in white. This graphical dependence once again reflects the complex nature of the nonlinear interaction. The time of the maximum wave in the entire set of experiments can vary from several minutes to a couple of tens of minutes for the same station. For example, the time of the wave maximum at station 18 in the experiment BE-3 is only a couple of minutes, and for the rest of the experiments, it varies from 14 to 20 min. This indicates that the waves in Palu Bay interact in such a way that their intensification occurs much later and with a larger amplitude than in BE-3.

Figure 6 shows the maximum wave height in the flood area of the model realisation (BE-3) and comparison with the observational data at the stations. For two stations located in the area of the city of Palu and two stations on the east coast, the time series of the wave height is given. At the given stations, the difference between the observed values and the model results remains small and does not exceed 30% of the maximum recorded by visual observations. Such a pretty good coincidence of the wave height is explained by the influence of only a single landslide on the flood zone. In other stations of the region, a complex picture of the interaction of waves occurs, and it is not easy to calculate for a better agreement with observations. A detailed analysis of the areas of influence of individual landslides and zones of strong nonlinearity is presented in the next section.

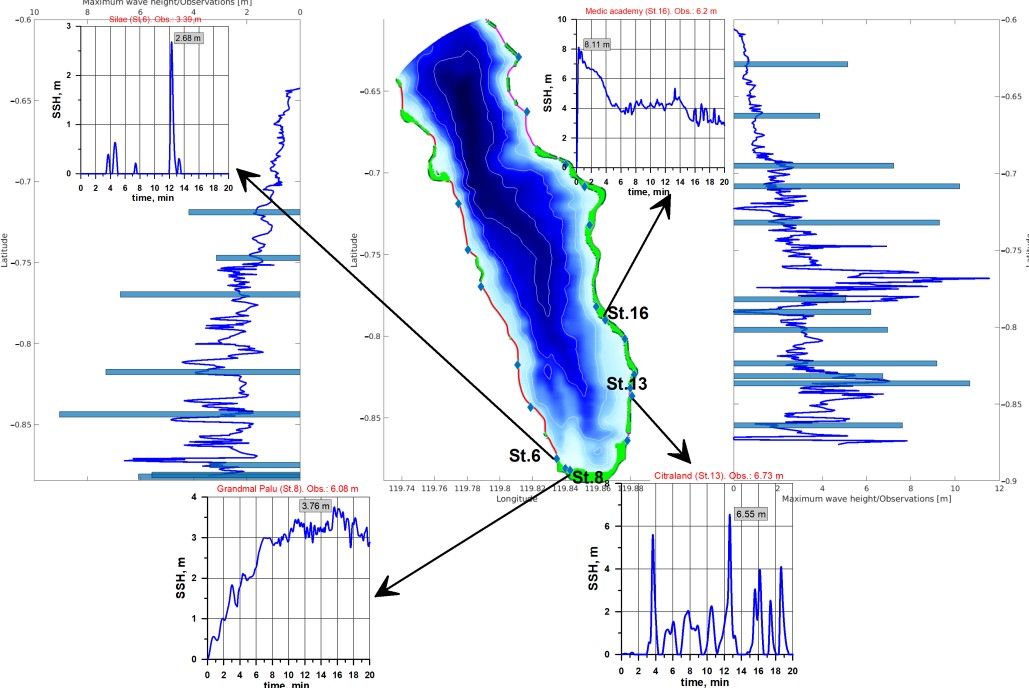

**Figure 6.** Comparison of the maximum wave height in the BE-3 experiment with observations [46]. The central panel shows the bathymetry of the simulated area and the observation stations. The flood zone in model calculations is highlighted in green. On the left and right (west and east coast) charts, the solid blue line shows the maximum wave height in the inundation area. The light blue bars show

the wave height according to the observational data. The four insets show the time series of sea level height at four points in the modelled area. According to observations, the name and maximum height are highlighted in red.

## 4. Analyse the Contribution of Individual Landslides to the Overall Dynamic Structure

Figure 7 shows the energy characteristics of the landslide-water system both in the computation of the joint movement of landslides and the movement of each landslide separately. From the comparison of kinetic energy, it can be seen that landslide energy predominates over the kinetic energy of the wave generated by it on average over the simulation period of 13.5 times. The average potential wave energy during the modelling period is approximately 4.3 times less than the kinetic energy.

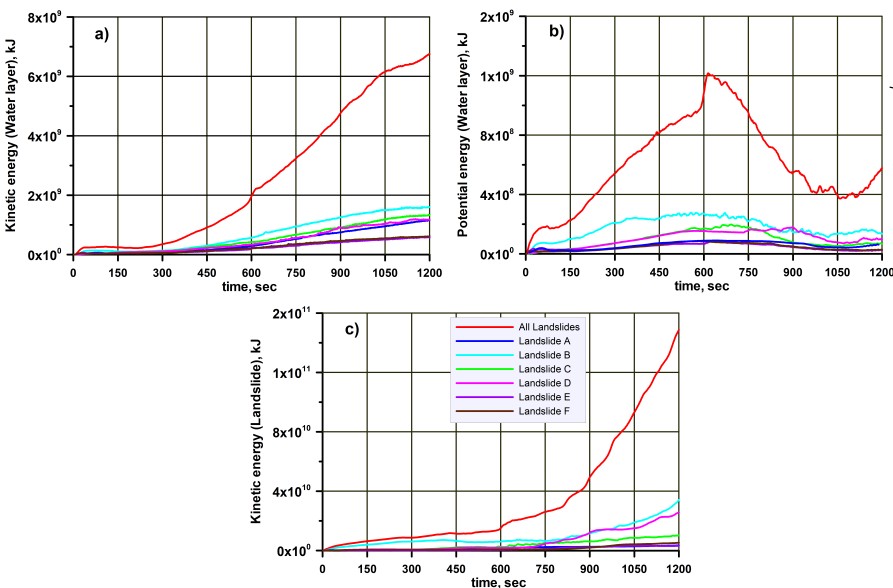

**Figure 7.** Kinetic and Potential energy. (**a**) Kinetic energy of the water layer; (**b**) potential energy of the water layer; (**c**) kinetic energy of the landslide.

Energy characteristics in the calculation of individual landslides show that landslide *B* plays a dominant role in the total formation of the tsunami wave. This landslide does not have maximum dimensions (see Table 1), but its position on the steeper slope forms additional acceleration at the initial moment. The second most crucial landslide in the energy system is landslide *D* with the largest size (Table 1).

Potential energy has its maximum at 600 s from the moment of formation of the maximum wave height. After that, the coastal zone is flooded and the potential energy decreases. A significant increase in the landslide kinetic energy after ∼600 s of calculation is explained by the fact that, at this moment, landslides reach the shelf edge and accelerate significantly on a sharp bathymetry in the central part of the bay. (Figure 2 left panel).

A comparison of energy characteristics makes it possible to evaluate the influence of each landslide and the degree of non-linear interaction in the modelling of landslides. The difference between the energy of the system of 6 landslides and the sum of the energies of a separate simulation of landslides is shown in Figure 8.

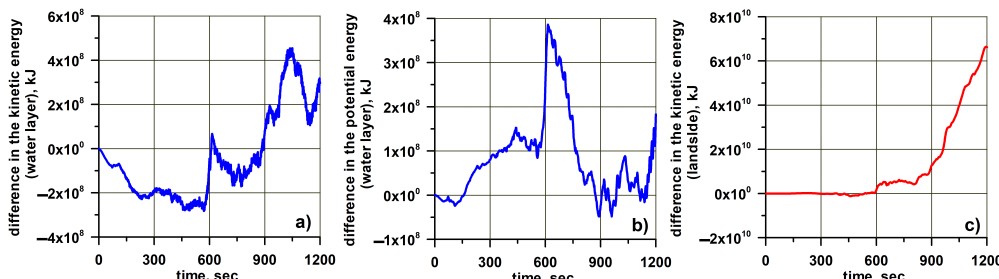

**Figure 8.** The difference in the energy between simulation for all landslides (BE-3) and summary energy for individual simulation landslides. (**a**) Difference in kinetic energy (water layer); (**b**) difference in potential energy (water layer); (**c**) difference in kinetic energy (landslide).

As seen from Figure 8c, the kinetic energy of a landslide has an almost linear movement character up to 800 s of model calculation. Further, the nonlinear nature of the movement of landslides prevails, and the contribution of this nonlinearity reaches a maximum of 30% concerning the maximum of total energy. The nonlinearity in potential energy is not so pronounced and does not exceed in maximum (by about 600 s of the estimated period) 6% (Figure 8b). The values for nonlinearity in the kinetic energy of the water layer are significantly higher and reach 40% in maximum (Figure 8a). In other words, non-linearity at the moment equation significantly dominates nonlinearity in the continuity equation by changing the thickness of the layer [38,39].

Another aspect of separate landslide modelling makes it possible to evaluate its contribution to the formation of wave height at observation points when compared with the calculation of total landslides. Figure 9 shows the correlation coefficient obtained from comparing the time series of calculating individual landslides and their joint movement. The high correlation coefficient between the separate calculation of the *B* landslide and the BE-3 experiment at stations 3 and 16 shows that the wave height here is formed under the action of the *B* landslide. Similar comparisons show that at stations 7–9, wave formation occurs under the influence of the *D* landslide. As can be seen from the results, a high correlation of the influence of an individual landslide on the wave formation occurs either at nearby stations to the landslide or, for example, at station 3, lying on the opposite side from landslide *B*, with weak interaction with counter waves from other landslides. At other stations, the wave formation process is of a complex nonlinear nature due to the interaction of waves from various sources.

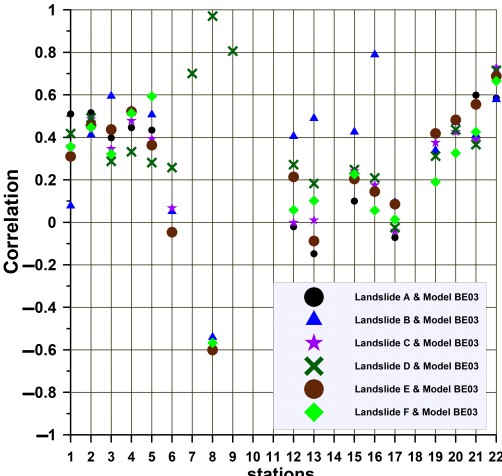

**Figure 9.** Correlation between simulation for total number landslides (BE-3) and computation for separate landslides.

## 5. Discussion and Conclusions

Against the background of the lack of complete information about bathymetry, topography and landslides, modelling the generation of tsunami waves and the flooding of coastal regions is challenging. Based on numerical modelling, we attempted to select the optimal characteristics of the landslides, the time of their initial movement and model parameters.

The complexity of the problem can be estimated by analyzing the spectra for the wave height solutions for one point (Pantoloan, Station 19, Figure 2 left panel) for several selected scenarios. For such an analysis, we use the Fourier spectral method, which is widely used in analysing tsunami wave frequencies [47]. The frequency spectrum is shown in Figure 10. As can be seen, the spectra of different scenarios vary significantly. The spectral pattern has one common element—intensifying the signal at frequencies of 4–6 min. Otherwise, the spectra contain pretty significant differences. For the scenario with a time delay in triggering landslides (BE-8, Table 2), a strong dominance of the high-frequency signal is observed for about 15 min. In the experiment (FE-4) with a reduced density of landslides, the frequencies of the minute range have a slightly higher amplitude (∼1.5 times) than in the main experiment (BE-3). An exciting picture arises when analyzing the frequency spectrum of a separate landslide *B*. The frequency spectrum, in this case, resembles the spectrum of the main experiment (BE-3), but the amplitudes of the oscillations are almost halved. These results reflect a complex picture of the interaction of waves generated by several landslides and reflected waves in the narrow Palu Bay.

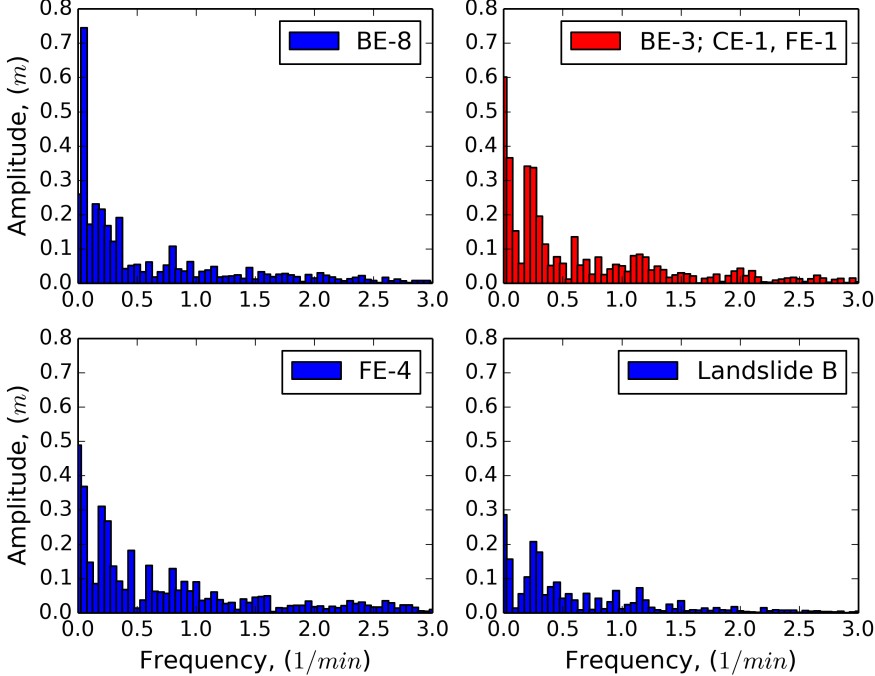

**Figure 10.** Frequency (1/min.) domain signal of the wave height from model simulations in the Pantoloan tide gauge (Station 19, Figure 2 left panel). **Upper left**: BE-8; **upper right**: BE-3; **bottom left**: FE-4; **bottom right**: individual landslide *B*.

Analysis of the frequency spectrum of the four selected scenarios shows significant differences in the amplitudes of the same frequencies (Figure 11). The most notable difference is at zero frequency (average elevation value). Thus, the average amplitude elevation difference in the main BE-3 experiment is more than twice that of experiment BE-8. In the high period of the spectrum (30–140 s), there is a substantial difference in the amplitudes of the wave spectrum in the analysed station. These differences again indicate a very complex nonlinear interaction of waves generated by landslides with different initialisation of the model.

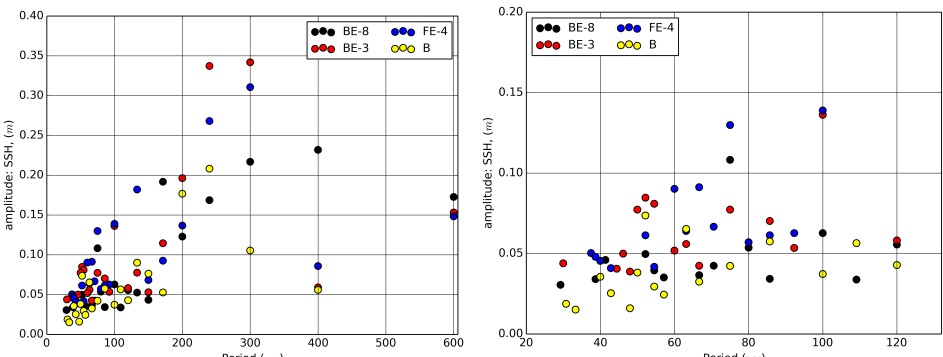

**Figure 11.** The first 20 maximum amplitudes of the frequency spectrum of 4 experiments. **Left** panel—entire period range; **right** panel—period range from 20 to 120 s.

In final, we will make some general conclusions about the presented work.

Given a large number of degrees of freedom in the problem being solved, a detailed study of the event depends on significantly better data coverage of the subject area, which includes more accurate bathymetry and landslide characterisation. In the absence of this information, it is necessary to substitute by selecting the missing information. It is shown how, in a series of experiments, from general scenarios to more localised ones, we attempted to select landslides and some model parameters based on a comparison with the available visual observations.

An analysis of the results shows that when modelling several landslides, sensitivity to their chronological and spatial order must be addressed due to the complex nonlinear interaction of the waves they cause. The complex picture of the interaction of waves generated by the landslide system significantly complicates the choice of the initial condition. Note that the density sensitivity (FE-1–FE-5, see Table 4) is less than the volume sensitivity (BE-10–BE-17, see Table 2).

The RMSD error for all visual observation points (22 points, Figure 2) ranges from 4.9 m to 7.4 m for the experiments. Once again, we note that all observation stations were located in the coastal zone, while some of the model points were in the coastal zone, which significantly distorted the quality of the comparison. When points outside the coast are excluded, the total error with observations is less.

An analysis of the water-landslide system's energy characteristics shows that a landslide's kinetic energy is more than an order of magnitude higher than the kinetic energy of the entire water column. The potential energy of the generated wave is almost eight times less than the kinetic one and has a maximum of approximately 600 s after the start of the landslide movement. Subsequently, due to the transformation of waves near the coastal zone, part of the potential energy is converted into kinetic energy, which leads to a significant intensification of horizontal velocities in the coastal zones.

During the analysis of model calculations, it was found that the effect of bottom friction is significant and requires an additional study on more detailed grids with bathymetry and high-resolution topography, as shown in [48]. We also note that the calculations did not consider the influence of a seismic source on the formation of a tsunami wave. Such an account would greatly complicate the choice of a scenario for waves formed under the influence of a landslide mechanism due to the lack of a seismic wave arrival time and its exact value at the model open boundary.

**Author Contributions:** Conceptualization, A.A., S.H. and N.R.; numerical model development, A.A.; numerical simulations, A.A.; data processing, A.A. and S.H.; writing the original draft, A.A.; review and editing of the manuscript, A.A., S.H. and N.R.; visualization, A.A. and S.H. All authors have read and agreed to the published version of the manuscript.

**Funding:** This research received no external funding.

**Data Availability Statement:** The data presented in this study are available on request from the corresponding author.

**Conflicts of Interest:** The authors declare no conflict of interest.

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
