# Peer review of "Simulating Landslide Generated Tsunamis in Palu Bay, Sulawesi, Indonesia"

_geosciences, doi:10.3390/geosciences13030072_

Round 1

Reviewer 1 Report

The manuscript titled “Simulating landslide generated tsunamis in Palu Bay, Sulawesi, Indonesia” by Androsov et al. reconstructed tsunami waves in Palu Bay under the influence of a landslide mechanism using the GNOM-LS model. It also analyzed the complex picture of the nonlinear interaction of tsunami waves triggered by the landslide. The best agreement with the observations was found through three experiments.

As a researcher in tsunami science, I agree that their work could be a great contribution to tsunami society. The generation mechanism of non-seismic tsunami (e.g., landslide tsunami) attracts more attention recently. Their method was well designed, and the manuscript was logically organized. It is suitable for publication in Geosciences. However, there are some problems with the “incorrect and coarse” bathymetry, and the method of spectral analysis was not clearly described. Hence, I believe a moderate revision was needed.

1)    Line 38: For landslide hydrodynamics model set by its center of gravity, the following paper on two-layer landslide tsunami model could be referred. This model was successfully applied to the landslide of the 2018 Anak Krakatau eruption in Indonesia.

https://doi.org/10.1007/s11069-020-03907-y

2)    Line 120: The horizonal viscosity was determined according to the Smagorinsky parameterization. How did you decide the value of coefficient As to guarantee its numerical stability?

3)    Line 185: In the final series of experiment (FE), why did you only change the parameter of landslide density rather than others (e.g., friction)?

4)    Table 1: The title of each column was not clear. “Slide volume” appeared in both the 2nd and the 3rd columns. Are there any differences? Please explain.

5)    Line 236: The authors acknowledged that the initial bathymetry was incorrect and coarse. It is necessary to discuss how large the discrepancies it would result in. Are those simulated results based on incorrect bathymetry reliable? How could we use those results to calculate the agreement with real observation? Similar questions will be raised in Lines 272–275, and Line 382.

6)    Figure 5: The color among the x-axis (experiment number) should be discrete, not continuous.

7)    Line 333: It should be mentioned that the high correlation coefficient is mainly due to the close location and propagation direction.

8)    Line 343: Here the authors should provide more details about spectral analysis. What type of spectral analysis did you use? It should also be noted that spectral analysis was widely applied to tsunamis inside a bay/harbor to study the resonance characteristics. The following papers could be referred:

https://doi.org/10.1093/gji/ggac291

https://doi.org/10.3390/jmse10081005 

9)    Line 350: The reviewer does not agree with the statement “… frequencies of the minute range of oscillations dominate relative to the main experiment”. It is not evident in Figure 10.

10) Figure 10: The spectra should be plotted in curve, instead of histogram. In addition, please mark the dominant period in “minute” unit.

Author Response

The authors are grateful to the reviewer for careful reading and correct comments. We hope that the article has become more informative.

Authors

Reviewer 2 Report

Summary

 A series of  numerical model experiments using the GNOM-LS, a landslide model using Navier-Stokes equations were conducted to model the tsunami on September 28, 2018, in Palu Bay using available observational data. Sensitivity analysis was done on the distribution of wave heights along the coastline with 22 stations, depending on the model parameters and the landslide characteristics. It is shown that the system has a high sensitivity to the initial conditions and the landslide characteristics and is highly non-linear. Based on the numerical modeling, the authors attempted to find the optimal characteristics of the landslides, model parameters, and the time of landslides’ initial movements.

 Comments: The manuscript content is relevant and clear with good experimental design. The numerical model is well described, and the results are presented well.

  • Given the large number of DOF in the selected problem, the modeling efforts are to be appreciated.
  • Since the observed surface elevation time series are available at stations and are presented, it is suggested to include a time obtained from the numerical model as well at least for two cases (least error and station with highest wave height recorded).
  • It would be good to substantiate the reasoning behind triggering the landslides only in the clockwise and anti-clockwise direction in contrast to other approaches such as radial distances or slopes.
  • From Fig. 5, stations 1 to 9, have good agreement with the observed data while stations 10 to 20 mostly underpredict the maximum wave heights with much larger errors.  The authors do well explain the uncertainties, However, since stations 10 to 20 are located on one side of the bay, is it possible/likely that the impact of the landslides F and E are underpredicted in terms of momentum or energy transfer based on the numerical results?
  • The authors provided sufficient information to substantiate the conclusions.

Author Response

We are grateful to the referee for carefully reading the manuscript, informative questions, and for an interesting suggestion to supplement the article's content.

Authors
